# Phytochemical Composition, Antimicrobial, Anticancer Properties, and Antioxidant Potential of Green Husk from Several Walnut Varieties (*Juglans regia* L.)

**DOI:** 10.3390/antiox12010052

**Published:** 2022-12-27

**Authors:** Sorour Barekat, Ali Nasirpour, Javad Keramat, Mohammad Dinari, Messaouda Meziane-Kaci, Cedric Paris, Stephane Desobry

**Affiliations:** 1Department of Food Science and Technology, College of Agriculture, Isfahan University of Technology, Isfahan 84156-83111, Iran; 2Department of Chemistry, Isfahan University of Technology, Isfahan 84156-83111, Iran; 3Institut Européen des Antioxydants (IEA), 54230 Neuves-Maisons, France; 4Laboratoire d’Ingénierie des Biomolécules, Université de Lorraine, 54500 Vandoeuvre-lès-Nancy, France

**Keywords:** antioxidant power, PAOT technology, phenolic compounds, green walnut husk

## Abstract

Husk powder was prepared from seven varieties of walnut fruit and their hulling rate, chemical compounds, and total phenolic contents were evaluated. The apolar and polar extracts were prepared, respectively, from hexane and a hydroethanolic solvent, while qualitative and semi-quantitative analyses were performed by GC/MS and UHPLC-PDA-HRMS/MS. The antioxidant, antimicrobial, and antitumor properties of green walnut husk were also evaluated. The total content of phenolic compounds varied between the varieties, ranging from 35.2 ± 0.9 to 58.0 ± 0.0 mg/g gallic acid equivalent of dry husk weight (dw). The apolar extract was found to contain alkanes, tocopherols, sterols, and fatty acids, including oleic, linoleic, and linolenic, while the polar extract showed the presence of phenolics including salicylate glucuronide, taxifolin, catechin, and quercetin isomers. The antioxidant power obtained by the PAOT (total antioxidant power) method for the husk powders ranged from 256.5 ± 5.9 to 746.8 ± 6.9 score/g dw, and seemed consistent with the total phenolic content and the results obtained by the classic antioxidant test with DPPH. The walnut husk also showed an antibacterial effect against Gram-negative and Gram-positive bacteria and cytotoxic potential against HepG2. Among the selected varieties, the green Saman had the highest antioxidant properties, while the Saman with a brown color had the lowest.

## 1. Introduction

Nowadays, due to safety concerns and growing consumer demand for clean-label food ingredients, the extraction of natural and low-cost antioxidants from the non-edible parts of fruits, vegetables, and herbs, including the peel, leaves, husks, and stems, has attracted considerable attention [1]. Walnut is a deciduous tree of the Juglans species belonging to the Juglandaceae family and is one of the most valuable agricultural products in the world. In walnut production, the kernel is the edible part of the fruit, and other parts such as green husks, septum, and bark are discarded as agricultural waste, while they are sources of valuable components [2]. The green husk is the major waste product, accounting for 45–60% of the fruit weight, depending mainly on the variety [3,4]. The vast amount of green husk produced during harvest and nut commercialization poses a major environmental problem that limits the development of fresh walnut production. Using the green husk is therefore a solution to protect the environment [4]. Phenolic compounds are secondary metabolites of plants and are considered antioxidants due to their ability to donate hydrogen (H) atoms to free radicals [5]. The walnut husk is a rich source of tocopherols, phenolic and flavonoid compounds with antioxidant properties [6]. However, previous studies have reported confusing results regarding the antioxidant activity of the husks [2,6,7,8,9]. 

Antioxidants, based on their mechanisms in neutralizing the adverse effects of oxidative stress, are divided into primary (scavenging), secondary (preventing), and tertiary (renovator, and repairer) antioxidants [5,10]. Scavenging antioxidants terminate oxidation reactions by donating hydrogen or electrons to free radicals and transforming them into more stable molecules. Assays to measure antioxidant activity, therefore, include single electron transfer reactions (FRAP, TEAC, and DPPH, etc.) and hydrogen atom transfer reactions (TRAP, ORAC, etc.) [10,11]. Most of these methods are simple and provide the possibility to compare results. However, obtaining non-same and non-repeatable results for a substance by one method and the irrelevance of the results from other different methods and laboratories are the drawbacks of these methods. Moreover, the values obtained by these methods cannot be compared with the in vivo results; consequently, to achieve reliable results, these methods must be combined with cellular methods, which are mostly time-consuming and expensive analyses [12].

PAOT liquid technology is a novel and non-destructive method and has the potential to directly measure the antioxidant capacity of food supplements (e.g., vitamins, extracts, minerals), foods (e.g., juice, tea, coffee), drugs, and biological fluids (e.g., blood and its fractions, seminal fluid). The PAOT technique determines the total antioxidant/oxidative activity based on the electrochemical properties of oxidative stress. In this method, two microelectrodes, a working electrode and a reference electrode, are immersed in the reaction environment containing a molecule in a free radical state called a mediator (M•) and after adding the sample, the changes in electrochemical potential which occurred due to changes in the concentration of the oxidized/reduced forms of the intermediate (M•) in the reaction with antioxidants is recorded [13,14,15]. Since the electrochemical method is associated with the electron–proton transfer and oxidative stress (the imbalance between antioxidants and pro-oxidants) which happen during the antioxidant action, it is a more robust and reliable approach than the previously mentioned spectrophotometric methods [16]. PAOT technology has many advantages, above all, by performing a simple automatized protocol in controlled environmental conditions similar to biological fluids and also recording electrochemical changes via the operator-independent device leads to repeatable and reliable results being achieved. In addition, there is no requirement for equipment such as a spectrophotometer, and results are recorded in a very short time [13,14,15]. 

In the current study, the anti-radical properties of walnut husk powder were evaluated for the first time. The chemical composition of the polar and non-polar extracts, total phenolic contents, antioxidant activities based on the reduction of DPPH free radicals, and by a novel electrochemical method, the antibacterial and anticancer properties of seven varieties of husks were evaluated. In addition to investigating the relationship between the presence of phytochemicals and antioxidant properties, the difference between green and brown dried walnut husk powder was determined. The aim was to assess the antioxidant capacity of the walnut husk as a new source of antioxidants using the PAOT method.

## 2. Materials and Methods

### 2.1. Materials 

The walnut fruits (*Juglans regia* L.) were collected from specific varieties (Kabiri farm, Saman region, Shahrekord, Iran) including Aytek, Dena, Chandler, Raha 3, Raha 4, Raha 6, and Saman, after the full ripening of the fruit from late August to September 2021. For each variety, fruits were harvested from 4 trees with a lifespan of 23 years and under the same geographical and agronomical management. Healthy fruits without walnut anthracnose infection or pathogen damage were selected. The fruits of selected varieties were different in terms of morphology, size, and percentage of green husk. Green husks were separated carefully by hand. Hulling rate, which represents the ripening stage of walnut fruits, was measured. Only the green husks that were completely separated from the walnut fruit were considered “hulling” and those green husks incompletely removed were considered “non-hulling”. Hulling rate was calculated according to the following Equation (1):Hulling rate % = (A_o_/A) × 100(1)
where A_o_ and A are the number of “hulling” and the total number of walnut fruits in 6 kg of walnut fruit, respectively, [17].

After that, the husks were immediately frozen, and completely freeze-dried (48 h) using a freeze dryer (Dena Vacuum, Iran). Only for Saman variety (sample with highest phenolic content and antioxidant properties), freeze-drying was carried out before and after discoloration from green to brown. To obtain the brown type of Saman, fresh husk was stored in the refrigerator (4 °C) for 2 weeks. The dry biomasses were milled and sieved to obtain a fine and homogeneous powder (size < 1 mm). Finally, the obtained powder for each variety was packed under nitrogen gas and kept at −80 °C prior to extraction. 

Ethanol, acetonitrile, formic acid, 2,2-diphenylpicrylhydrazyl (DPPH), Folin & Ciocalteu’s phenol reagent, *Staphylococcus aureus* (ATCC 25923), *Bacillus subtilis* (ATCC 6051), *Escherichia coli* (ATCC 25922), *Pseudomonas aeruginosa* (ATCC 27853), and all bacteria media were purchased from Sigma-Aldrich Co. (St. Louis, MO, USA). Other chemicals and solvents were analytical grade and purchased from Merck Co. (Darmstadt, Germany). For all analyses, bi-distilled and purified water was prepared using a Milli-Q system (Merck Millipore, Burlington, MA, USA). 

### 2.2. Chemical Composition of Walnut Husk Powder

Moisture, ash, and fat content of husk powder samples were measured according to the AOAC standard methods as 968.11, 942.05, and 948.22, respectively. Total carbohydrate was determined by rapid colorimetric phenol-sulfuric acid assay; d-glucose was used as standard at 490 nm [18]. Content of protein was determined using the micro Kjeldahl method with a conversion factor of 6.25, equivalent to 0.16 g nitrogen per gram of protein [19]. Inductively coupled plasma-optical emission spectrometry (ICP-OES; Optima 7300 DV, Perkin Elmer, Shelton, CT, USA) was used to analyze the main elements. About 5 g of each sample was weighed into dry vessels and heated in an electric oven at 550 °C until the heavy evolution ceased. The final residues were digested in a hot water bath with freshly made HNO3 (5%). To assess the mineral content of sample solutions, they were filtered and then introduced into the ICP-OES [20].

### 2.3. Preparation of Apolar and Polar Extracts for Chemical Analysis

For phytochemical analysis, the husk powders were first extracted by hexane (non-polar extract) and then air-dried plant residues were extracted by ethanol and bi-distilled water (polar extract). The extraction process steps of polar and apolar compounds were the same, and powder to solvent ratio was 1:20 (both for hexane and ethanol 80%). Briefly, the mixtures of plant material and solvent were first shaken well for 5 min using a vortex mixer (TOP-MIX 94,500; Heidolph, Schwabach, Germany), and then sonicated in iced water (Fisherbrand S-Series, ThermoFisher, Milan, Italy) for 60 min. Centrifugation (Z36 HK, Hermle, Germany) at 13,540× *g* for 10 min was performed to accelerate the extraction of compounds. For polar compounds, a 0.2 µm polyamide filter (Chromafil AO-20/25; Macherey-Nagel, Düren, Germany), and for apolar compounds, a 0.2 µm polytetrafluoroethylene filter (Chromafil Xtra PTFE-20/25 Macherey-Nagel, Duren, Germany) was used. The resultant polar and non-polar extracts were then transferred to vials and stored at −32 °C until further analysis [21]. 2.4. Qualitative Analysis of Apolar Extract by GC-MS

### 2.4. Qualitative Analysis of Apolar Extract by GC-MS

The GC-MS analysis of non-polar extracts was performed using a GC-MS system (Agilent Technologies 5973N GC/MS System, Santa Clara, CO, USA). A 5% Phenyl Polysilphenylene-siloxane (BPX5) column (30 m length, 0.25 mm internal diameter, and 0.25 μm film thickness) was used for separation. The temperature programming was set in the range 50–270 °C, by holding 4 min at 50 °C, then reaching 270 °C with an increasing rate of 2 °C/min and finally holding for 15 min at 270 °C. One microliter of the non-polar extract was injected in 1:10 split mode. Ultra-pure helium was used as carrier gas. For identification, the peak retention time (RT) and the experimental electronic impact mass spectra (EI-MS) were compared with EI-MS of the Wiley digital library [17].

### 2.5. Fatty Acid Profiles 

Methylation of fatty acids of apolar husk extracts was performed according to the method of Goli et al. [22]. Briefly, 100 μL of 0.5 M sodium methoxide was added to 1 mL of apolar extract in hexane. The mixture was shaken vigorously by a rotary shaker for 20 min and allowed to stand. Then, 5 μL of the top phase was analyzed using the GC instrument (6890 N, Agilent Technology Inc., Palo Alto, CA, USA) equipped with a HP-88 (100 m, 25 mm, and 0.2 μm film thickness) column. Temperatures of column and detector were set at 185 °C and 240 °C, respectively. The amount of each fatty acid was expressed relative to the total fatty acid content (semi-quantitation based on area ratio).

### 2.6. Qualitative and Semi-Quantitative Analysis of Polar Extract by UHPLC-PDA-HRMS/MS

The extracts of green walnut husk were analyzed on a ThermoVanquish quaternary UHPLC system in-line with a photodiode array detector (PDA) and a ThermoScientific Orbitrap ID-X Tribrid mass spectrometer equipped with an atmospheric pressure ionization interface operating in electrospray mode (ESI). The measured 10 µL samples of extracts were separated on a Hichrom Alltima C18 column (150 × 2.1 mm−5 µm) maintained at 25 °C. The flow rate was set at 200 µL min^−1^ and mobile phases consisted of water modified with the formic acid (0.1%) for A and acetonitrile modified with the formic acid (0.1%) for B. Compounds of interest were eluted using a linear gradient from 5% to 95% of B for 50 min, then an isocratic step was applied at 95% B for 5 min to wash the column, before returning to the initial composition of 5% B for 5 min to realize the equilibrium. Mass analysis was carried out in heated-electrospray positive ion mode (HESI^+^) and mass spectrometry conditions were as follows: spray voltage was set at +3.5 kV; source gases were set (in arbitrary units. min^−1^) for sheath gas, auxiliary gas and sweep gas at 35, 7 and 10, respectively; vaporizer temperature and ion transfer tube temperature were both set at 300 °C. 

Survey scans of precursors were performed from 150 to 2000 *m*/*z* at 60 K resolution (full width of the peak at its half maximum, FWHM, at 200 *m*/*z*) with MS parameters as follows: RF-lens, 35%; maximum injection time, 50 ms; data type, profile; internal mass calibration EASY-IC TM activated; automatic gain control (AGC) target: 100,000; normalized AGC target: 25%. A top-speed data-dependent MS2 was performed using quadrupole isolation (1.5 Th window), HCD fragmentation with stepped collision energy (25, 35, and 50%), and an orbitrap analyzer at 15 K resolution (high-resolution MS/MS analysis). Only precursors with an intensity above a threshold of 2.104 were selected for MS2. Dynamic exclusion duration was set to 2.5 s with 10 ppm tolerance around the selected monoisotopic signal (isotopes excluded). Other MS2 parameters were as follows: data type, profile; AGC target: 10,000; normalized AGC target: 25%. 

Raw MS and MS2 data were then submitted to the software Compound Discoverer 3.3 to perform automatic detection and identification of compounds present in green walnut husk extracts. The identification process involved querying three different databases, namely (i) the efficient “mzCloud” (ThermoScientific) based on raw formulas (by exploiting the isotopic pattern of the parent ion) and high-resolution experimental MS2 spectra, (ii) “ChemSpider” (Royal Society of Chemistry) based on raw formulas of a large batch of biomolecules and their high-resolution in silico MS2 spectra and (iii) “Metabolika” (ThermoScientific) based on the raw formulas of biomolecules extracted from more than 370 biochemical pathways organized and annotated for a range of organisms. When compounds of interest were not automatically identified, manual processing was applied by interpreting characteristic daughter ions and neutral losses. 

The mass spectrometer calibration was performed using the ThermoScientific Pierce TM FlexMix TM calibration solution. MS data acquisition was carried out using the Xcalibur v. 3.0 software (Thermo Fisher Scientific, Waltham, MA, USA).

### 2.7. Total Phenolic Content

The total phenolic content of polar extracts of walnut husk was measured by spectrophotometric based on the Folin–Ciocalteu method. Briefly, 0.1 mL of sample, standard (gallic acid solutions from 0 to 500 ppm) or blank (solvent), was mixed with 0.2 mL of Folin–Ciocalteu’s phenol reagent and 2 mL of water for 3 min., before addition of 1 mL of 20% sodium carbonate solution. The mixtures were incubated for 1 h at 25 °C and the absorbance of the resultant blue color was measured at 765 nm using a spectrophotometer (PowerWave XS, BioTek, Inc., Winooski, VT, USA). Finally, the total phenolic contents were expressed as mg equivalent of gallic acid/g of dry weight (dw) of husk powder [7].

### 2.8. Total Antioxidant Activity by DPPH Assay

The DPPH assay was used as a common method for determining the antioxidant activity of polar husk extracts. Briefly, 150 mL of 11 different concentrations (1000–0.97 µg/mL) of the samples were mixed with 150 µL of methanolic solution of DPPH (0.04 mg/mL). After 30 min incubation at 37 °C, absorbance was measured at 517 nm by UV Spectrophotometer (PowerWave XS, BioTek, Inc., Winooski, VT, USA) and used for calculation of half-maximal inhibitory concentration (IC50).
(2)$$Inhibition(\%)=(1-\frac{Abs_{sample}-Abs_{blank}}{Abs_{control}-Abs_{blank}})\times 100$$
where *Abs* sample was the absorbance of the reaction medium in presence of sample (sample dilution + DPPH solution), *Abs* blank was the absorbance of the blank for each sample dilution (sample dilution+ DPPH solvent) and *Abs* control was the absorbance of control reaction (sample solvent +DPPH solution).

### 2.9. Total Antioxidant Power by PAOT Technology

The antioxidant power of walnut husk extracts was measured using the total antioxidant power assay (PAOT Technology, European Antioxidant Institute, Nancy, France). The measurement was carried out in a liquid medium by solubilizing the sample in the reaction media containing a free radical molecule called intermediate (M•), and the antioxidant activity was then assessed directly in the solution. In PAOT technology, which mimics biological circumstances by using physiological solutions with a pH range of 6.7 to 7.2, samples were combined with 2.5 mL of the reaction medium. At a room temperature of 25 ± 2 °C, two specific microelectrodes, one working electrode and one reference electrode, were immersed in the solution for a short period of time. After adding 20 µL of the sample (1 mM), the antioxidant power was determined by recording changes of the electrochemical potential in the reaction medium during the reaction (due to changes in the concentration of oxidized/reduced forms of the mediator M•). PAOT score (total antioxidant power) per gram of dw walnut husk was used to express the results. To compare the antioxidant power of the samples with food ingredients, their PAOT score scales were also presented [13,14].

### 2.10. In Vitro Antimicrobial Analysis by Broth Microdilution Protocol

The antimicrobial activity of the husk extracts (mixture of polar and apolar) was tested using a serial dilution titration method, to determine the minimal inhibitory concentration (MIC) and minimal bactericidal concentration (MBC). The bacteria collections (*Staphylococcus aureus*, *Bacillus subtilis*, *Escherichia coli*, and *Pseudomonas aeruginosa*) with an initial concentration of 40 mg/mL grew overnight at 37 °C in Mueller–Hinton Broth (MHB) and were diluted in the same medium. Two-fold serial dilutions of the husk extract were added to the micro-titer plates in a volume of 100 μL, followed by the addition of 100 μL of the bacteria solution to give a final inoculum of 5 × 105 colony-forming units (CFU)/mL. The plates were incubated at 37 °C for 24 h, and the MICs were determined. Finally, 100 µL of the 24 h inhibitory concentration test sample (MIC well) and its further concentrations were plated on Mueller–Hinton agar (MHA) and incubated at 37 °C overnight to determine MBC. The antibiotics penicillin and streptomycin were used as positive controls [23]. 

### 2.11. Anticancer Activity Analysis by MTT Assay

The human liver tumor cancer cell line (HepG2) isolated from a hepatocellular carcinoma of a 15-year-old, white, male youth with liver cancer (HB-8065) was cultured at 1 × 105 cells/well in 96-well plates for 24 h at optimal conditions (37 °C, 5% CO_2_ in humidified incubator). The growth media (10% fetal bovine serum (FBS)) was removed, and the cells were washed twice with PBS. New maintenance Roswell Park Memorial Institute (RPMI) medium (10% FBS), containing 0.05, 0.5, 5, 50, and 500 µg/mL of husk extract (mixture of polar and apolar), was added and the cells were incubated 24, 48, and 72 h. Triple wells were analyzed for each concentration and column elution buffer was used as the control. A 10 μL solution of freshly prepared 5 mg/mL MTT (3-(4,5-dimethylthiazol-2-yl)-2,5-diphenyltetrazolium bromide) in PBS was added to each well and allowed to incubate for an additional 4 h. The media was removed, and dimethyl sulfoxide (DMSO) was added at 100 µL/well. Plates were shaken gently to facilitate formazan crystal solubilization. The absorbance was measured at 545 nm using a microplate reader (STAT FAX 2100, BioTek, Winooski, VT, USA). The percentages of cell toxicity and half-maximal inhibitory concentration (IC50) were calculated [24,25].

### 2.12. Statistical Analysis

All the experiments were performed in triplicate. Analysis of variance (ANOVA) was performed on the data using the general linear model of Statistical Analysis System (SAS) version 13.1. The confidence interval was set for a level of significance at *p* < 0.05. 

## 3. Results and Discussion

### 3.1. Hulling Rate

The hulling rate of the green husk from the walnut fruit of different varieties was between 80.2 ± 0.1 and 90.0 ± 0.1% of fruits. In fact, the number of green husks completely separated in a certain weight of fruit is known as an indicator of the degree of post-harvest ripening and dehiscence. The hulling rate of the samples was over 80%, which shows the full maturation of the fruits. The ripening stage of the walnut fruit has an effect on the chemical composition and antioxidant properties of the green husk [17]. Therefore, in order to compare various varieties of fruits, they were selected only at the stage of full maturity.

### 3.2. Chemical Analysis 

The chemical composition of the green walnut husk powders was measured as carbohydrate, protein, fat, ash, and non-nitrogen content at 42.3–49.8%, 15.4–16.9%, 6.3–6.9%, 17.8–19.6%, and 8.6–16.4% based on dry weight, respectively. About 50% of the green husk of the walnut was composed of carbohydrates. Analysis of the metal ion content revealed that Ca was the highest with an average concentration between 9000 and 12,500 mg/kg of dried husk. The values of K, Na, and Mg in the green walnut husk powders were approximately 4500–4800 mg/kg, 140–159 mg/kg, and 1000–5500 mg/kg, respectively. In general, the percentages of chemical compounds and the content of metals present in plants depend on the variety. Similar results were reported by Soto-Maldonado et al. [26].

### 3.3. Analysis of Non-Polar Extract by GC-MS

As listed in Table 1, alkanes, alkenes, naphthoquinones, terpenes, sterols, and fatty acids were the main compounds. The most prevailing compounds were γ-sitosterol (2.47–33.8%), vitamin E (7.57–15.98%), β-tocopherol (1.43–2.92%), lupeol (1.04–2.28%), juglone (0.12–3.66%), and fatty acids (oleic acid, linoleic acid, palmitic acid, stearic acid); the other compounds always being present at less than 2%.

In walnut husk extracts, alkanes with different chain lengths (n-C9 to n-C28) were identified (Table 1). Short-chain alkanes typically originate from algae and bacteria [27], but they were also found in husk non-polar extract (Table 1). Seabra et al. [4] identified nine alkanes (tetradecane, tetracontane, docosane, pentacosane, pentatriacontane, heptacosane, eicosane, heptatriacontane, heneicosane) in the husk high-pressure supercritical extracts. However, in our study, more components, especially short-chain alkanes (heptane, octane, nonane, decane, undecane, etc.), were detected. Alkenes are another important type of hydrocarbons and the 1,19-eicosadiene was actually detected in non-polar extract. The presence of 1,19-eicosadiene was also reported by Seabra et al. [4]. The 1,4-naphthalenedione, 5-hydroxy- or juglone is a strong antioxidant and the main compound found in all parts of the walnut fruit [28].

The terpene compounds such as dl-limonene and β-limonene are the characteristic aroma components of walnut fruits and were detected in the husks [17]. Tocopherols have antioxidant activities and play a key role in the oxidative reactions of unsaturated fatty acids in foods and biological systems. α-Tocopherol and γ-tocopherol are the main vitamin E components [29,30]. Two tocopherol homologs (α- and γ-tocopherol) were found in the extracts. These compounds are liposoluble metabolites that are found in many plants [29]. The γ-sitosterol and lupeol are sterols with strong antifungal, antibacterial, and antispasmodic activities that were detected. In addition, different fatty acids and other compounds were found. In summary, the non-polar extract was mainly composed of volatile compounds which are probably related to the odor of walnut [17].

### 3.4. GC and GC-MS Fatty Acid Profiles

GC and GC-MS analysis was employed to identify the saturated and unsaturated fatty acids as follows: myristic (C14:0), palmitic (C16:0), and stearic (C18:0) as the saturated fatty acids and oleic (C18:1), linoleic (C18:2), linolenic (C18:3), and 11-eicosenoic (C20:1) as the unsaturated fatty acids. The fatty acid profile indicates a high unsaturated/saturated ratio and a high amount of oleic and linoleic acids (Table 2). In walnut kernels, the main fatty acids are palmitic, stearic, oleic, linoleic, and linolenic [31]. The results confirm that, compositionally, the fatty acids of green walnut husks were similar to the walnut kernel. In all the investigated varieties, oleic, palmitic, and linoleic had the highest percentages and eicosanoic acid was only present in Aytak and Raha 3. The comparison of the green and brown varieties of Saman showed an interesting result. The percentage of linolenic and linoleic unsaturated acids was significantly lower in brown Saman than in green Saman. The reason for these lower values in brown Saman could be explained in this case by the oxidation of unsaturated fatty acids.

### 3.5. Analysis of Polar Extract by UHPLC-PDA-HRMS/MS

The green walnut husk hydroethanolic extracts were analyzed by UHPLC-PDA-HRMS/MS, and their UV282 chromatograms are shown in Figure 1. Twelve of the main compounds observed in UV282 were identified thanks to their exact masses (parent ions seen in mono- or diprotonated forms) and their high-resolution MS2 fragmentation patterns. In detail, five of the twelve compounds of interest (namely minoxidil, myricitrin, quercetin 4’-glucoside, taxifolin and quercetin pentoside) were identified by automatic search in the high-resolution mass spectral library “mzCloud”. In addition, three compounds (namely catechin, abscisic acid and salicylate glucuronide) were automatically identified thanks to the “Chemspider” mass database. Finally, three other compounds of interest (namely neochlorogenic acid, taxifolin 7-glucoside and gallic acid) were identified manually because they were not found in the various databases queried. 

The name, formula, calculated molecular weight (Cal. MW), exact *m*/*z* values of parent ions, and retention time (RT) are shown in Table 3. To the best of our knowledge, four compounds are reported for the first time: taxifolin 7-glucoside, minoxidil, abscisic acid, and salicylate glucuronide, although others have been previously identified in walnut waste [21,28,32].

Flavonoids, as the largest group of natural polyphenolic compounds, exist in free or glycoside forms. The basal backbone of flavonoids is constituted of a benzene ring A linked to a gamma-benzopyrone (an oxidized heterocyclic C ring substituted in C2 by benzene B ring) by a three carbon bridge. Based on the location of the unsaturated bonds or substitution and the number of OH groups in the gamma-benzopyrone structure, flavonoids are divided into flavon, flavanol, flavonol, and flavanonol subgroups [39]. Cianidanol (*m*/*z* 291.09, RT = 11.77 min) is an antioxidant plant metabolite and belongs to the flavanol (flavanol-3-ol) family. Cianidanol or (+)-catechin is the (+)-enantiomer of catechin. In flavanols, the ketone group in the ring C (C4) does not exist. They have two OH groups in positions 5 and 7 in the benzene A ring, one in position 3 in the heterocycle C ring, and two in positions 4’ and 5’ in the B ring. The daughter ion at *m*/*z* = 139 indicates the presence of an A ring in the cianidanol structure [13,40]. 

Myricetin (*m*/*z* = 465.10, RT = 17.01 min), quercetin-4’-glucoside (*m*/*z* = 465.10, RT = 17.18 min) and quercetin-3-O-pentoside (*m*/*z* = 435.09, RT = 18.22 min) are flavonols. In this family, two OH groups in positions 5 and 7 of the benzene A ring and one in position 4 in the C ring are present [39]. For myricetin, three OH groups (positions 3’, 4’ and 5’) are substituted in ring B. Quercetin 4’-glucoside is quercetin (two OH groups in positions 3’, 4’ in ring B) with a beta-D-glucosyl residue attached in position 4’ [13]. Its intense daughter ion at *m*/*z* = 303.0 [M+H-C_6_H_12_O_6_] indicates the loss of glucose, and the presence of the ion moiety of quercetin [33]. Furthermore, the neutral loss of 133 (daughter ion [M+H-133]^+^) is linked to the departure of the pentoside moiety from the quercetin-3-O-pentoside. This compound was also reported in previous works in the septum of walnut [38].

Taxifolin (*m*/*z* = 305.07, RT = 17.40 min) or 5,7,3′,4′-flavan-on-ol belongs to the flavanonols. The daughter ion at *m*/*z* = 287.1 reveals the loss of a water molecule as reported by Sheng et al. [33]. Taxifolin is the dehydrogenated form of quercetin, and due to the absence of a double bond on carbon 2 and 3 in ring B, it has less antioxidant properties than quercetin [41]. Thanks to the daughter ion [M+H-C_6_H_12_O_6_] seen at *m*/*z* = 305.1, the compound associated to peak 2 was clearly identified as taxifolin 7-glucoside. The peaks 1 and 11 could be identified as neochlorogenic acid (5-O-caffeoylquinic acid) and gallic acid derivative, respectively. Neochlorogenic acid is a natural phenolic acid and is structurally constituted of a cyclitol carboxylic acid and a cinnamate ester. Gallic acid is a trihydroxybenzoic acid and hydroxy groups are present in positions 3, 4, and 5. These compounds were identified through their parent ions reported in walnut husk by Medic et al. [28] and Sheng et al. [33]. 1-Salicylate glucuronide (*m*/*z* = 315.07, RT = 15.99 min), a glucuronide conjugate of salicylic acid, is also a phenolic acid identified in husk extracts. 

The minoxidil (*m*/*z* = 210.14, RT = 13.30 min) structure is a pyrimidine-2,4-diamine 3-oxide that is substituted by a piperidin-1-yl group at position 6. Well known as a vasodilator, hair-growth stimulator and antihypertensive agent [42], this compound was identified for the first time in walnut husk, thanks to our work. (±)-(2E)-Abscisic acid or dormin (*m*/*z* = 265.14, RT = 13.34 min) is an abscission-accelerating plant growth substance and is found in different parts of plants [43].

The semi-quantitative analysis of the compounds in green walnut husk extracts was evaluated. Most of the identified compounds exist in all walnut husk extracts. The Saman variety, which was dried after storage and was brown in color, showed the greatest differences from the other samples and did not contain taxifolin 7-glucoside, or quercetin pentoside. In this sample, most compounds including catechin, taxifolin, and minoxidil were at their minimum level. Taxifolin and catechin were the highest in Reha 4; abscisic acid and salicylate glucuronide in the brown Saman sample and salicylate glucuronide and catechin were the highest in other samples (Figure 1). 

### 3.6. Total Phenolic Content

The total phenolic content of the dry husk powders of Aytek, Dena, Chandler, Raha 3, Raha 4, Raha 6, green Saman and brown Saman, was 56.1 ^c^ ± 0.0, 52.1 ^e^ ± 0.1, 45.0 ^f^ ± 0.1, 58.0 ^b^ ± 0.0, 55.6 ^d^ ± 0.2, 56.2 ^c^ ± 0.1, 59.8 ^a^ ± 0.2, and 35.2 ^g^ ± 0.9 mg GAE (gallic acid equivalent)/g dw of husk, respectively. Phenolics represent about 3.5 to 6% of the dry weight of husk powder, which means that this product could be known as one of the most phenolic-rich natural substances. Among the investigated varieties, Saman had the highest content of phenolic compounds in its green form and the lowest in its brown form. In previous works, the total phenolic content of green walnut husk has been reported as mostly dependent on the extraction method and on the type of variety. In a study by Rusu et al. [44], the phenolic content of Romanian walnut septum was found to be 67.03 ± 9.76 by the Ultra-Turrax method and 31.27 ± 5.24 mg GAE/g dw by maceration extraction. Oliveira et al. [7] reported that the phenolic content in the methanolic extract of husk ranged from 32.61 for the Mellanaise cultivar to 74.08 mg GAE/g dw for Franquette.

### 3.7. Total Antioxidant Activity by DPPH Assay

As can be seen in Figure 2a, the DPPH scavenging activity logically increases with the concentration of husk powder. As DPPH IC50 is the concentration required for 50% inhibition of DPPH free radical, therefore, lower values indicate higher antioxidant activity. The IC50 of samples was in the range 146.8–249.3 µg/mL of husk. These results are in accordance with the total phenolic content previously measured in each sample. The green husk varieties had different antioxidant activities, green Saman and Chandler had the highest and lowest, respectively. Furthermore, the Saman sample was examined in two colors, green and brown, and a significant difference in IC50 was observed depending on the color. The results obtained in the DPPH assay were always related to the peak intensity of the phenolic compounds. Oliveira et al. [7] measured the EC50 in the aqueous walnut husk extracts of different cultivars and the values were found between 350 and 590 µg/mL. Similarly, the EC50 of the walnut husk extracted by different solvents was reported in the range 330–720 µg/mL [8]. In another study, the obtained IC50 values of methanolic extracts, depending on the cultivar, ranged from 141 to 2890 μg/mL [45]. As mentioned before, the most important problem encountered when using common methods such as DPPH is the impossibility to accurately compare the results of different studies with each other. Therefore, in the following, the PAOT technology was used to reliably measure the antioxidant power.

### 3.8. Total Antioxidant Power by PAOT Technology

Figure 3 shows the PAOT liquid technology scale for ingredients and the total antioxidant activity of various varieties of dried green walnut husk powder. According to the PAOT scores, ingredient samples with a value below 250 are not labeled as antioxidants and samples with a value greater than 1000 are stated as very effective products in the same way as the reference molecules (Figure 3a). The average PAOT score/g dw of husk samples was in the range 256.52 ± 5.89 to 746.88 ± 6.96 (Figure 3b). According to the PAOT liquid technology scale for ingredients, two varieties were rated as effective, and five were rated as moderately effective. The brown variant of Saman was labeled as a product with weak effectiveness. It should be noted that these results are based on the dry powder and without any purification or isolation steps, so the green walnut husk can certainly be categorized as a strong antioxidant.

In the case of the green-colored powders, Saman showed the greatest antioxidant power (746.88 ± 6.96), while Chandler was the weakest. Between the two types of Saman, the brown showed a low antioxidant power compared to the green (about 40% less). This result could be due to the oxidation mechanism. Given that the only distinction between green and brown Saman powder was the length of storage, probably the freezing and quick drying of the sample are crucial to obtain a product with strong antioxidant properties. The antioxidant power was consistent with the results obtained for the total phenolic compounds and DPPH assay (except for Aytak). In samples with a high phenolic content, the antioxidant power was high, as expected. In addition, the chemical composition was probably another main reason for the difference in the antioxidant power of the extracts. In the green Saman sample, the peak areas of the antioxidant compounds were the highest, and in the brown-colored Saman, lower peak areas were observed. In brown Saman, the amount of abscisic acid, which has no antioxidant activity, had the highest peak intensity, which is probably the reason for its low PAOT score. Additionally, in other samples, the relationship between the peak intensity of the phenolic compounds and the antioxidant power was obvious. According to Figure 1, there was a clear difference in the peak area of catechin (with antioxidant capacity), which was very low in Saman brown. The differences in peak area observed for salicylate glucuronide, quercetin 4′-glucoside, quercetin pentoside and taxifolin 7-glucoside most certainly explain the differences in the measured antioxidant power. The flavonoid structure, with an OH group on the 3′ and 4′ carbons in ring B and an OH group in ring C, is probably crucial for inhibiting free radicals. Due to the presence of hydroxyl groups in ring A, the antioxidant property and free radical inhibition increase significantly. The double bond in carbon 2 and 3, which is conjugated with the carbonyl group, is responsible for electron distribution in the B ring and is the factor of increasing the antioxidant property. Taxifolin lacks a double bond on carbon 2 and 3, so its antioxidant property is lower than quercetin. The presence of a hydroxyl group on carbon 3 in ring C increases the antioxidant property. Therefore, glycosylation in this position reduces the antioxidant property [13,39,41]. 

### 3.9. In Vitro Antimicrobial Properties

The antibacterial properties of the husk extracts and positive controls (streptomycin and penicillin) are presented in Table 4. A ratio MBC/MIC (minimum bactericidal concentration/minimum inhibitory concentration) less than or equal to four is considered bactericidal, while a ratio MBC/MIC greater than four is considered bacteriostatic. Since the MIC to MBC ratio (1–2) for all bacterial collections was less than 4, all extracts of walnut husk exhibit bactericidal activity. The lowest and highest MIC were obtained for *E. coli* (0.5–1 mg/mL) and *P. aeruginosa* (4–7 mg/mL), respectively. The antibacterial properties of the extracts were probably related to the compounds in the husk of walnut. In the non-polar part, compounds such as monoterpenes, tocopherols, sterols and juglone and in the polar part, phenolic acids, have strong antipyretic, analgesic, anti-inflammatory, antimicrobial, and antioxidant properties [4]. 

The antibacterial activity of walnut husks was also investigated in previous studies and different results were reported. Vieira et al. [23] evaluated the antimicrobial properties of the hydroethanolic extract of walnut husks on Gram-negative (*E. coli*, *K. pneumoniae*, *M. morganii*, *P. mirabilis*, and *P. aeruginosa*) and Gram-positive (*E. fecalis*, *L. monocytogenes*, and methicillin-resistant *S aureus*) bacteria. The MIC of husk extract was 5 mg/mL for *S. aureus*, 10 mg/mL for *L. monocytogenes*, and *E. coli* and 20 mg/mL for *K. pneumoniae*, *M. morganii*, *P. mirabilis*, and *E. fecalis*. The extract showed no inhibition effect on *P. aeruginosa* (MIC > 20 mg/mL) [23]. Fernández-Agulló et al. [8] also reported antibacterial potential against *B. cereus* (20 mg/mL), and a weak potential against *P. aeruginosa* and *E. coli* (100 mg/mL). In the study of Oliveira et al. [7], the amounts of MIC against *S. aureus* (0.1 mg/mL), *Bacillus cereus* (0.1–1 mg/mL), *Bacillus subtilis* (0.1–10 mg/mL) and *P. aeruginosa* (100 mg/mL) for different Portuguese cultivars of green walnut husks were lower, which showed higher antibacterial properties. In general, it can be stated that the selected variety, the agricultural conditions, and the protocol used for the antibacterial measurement have an effect on the antibacterial properties of the extract of green walnut husk [23]. As is clear in Table 4, all the varieties had good bacterial characteristics, even a bactericidal effect on *P. Aeruginosa* was seen.

### 3.10. Anticancer Activity

Since the green Saman variety had the highest total phenolic, antioxidant, and antibacterial properties in all the analyses, the anticancer activity of its extract at different concentrations (0.05, 0.5, 5, 50, and 500 µg/mL) against the human liver tumor cancer cell line (HepG2) over 24, 48, and 72 h was investigated; the results are shown in Figure 4. The toxicity effect of walnut husk extract on HepG2 was dependent on concentration and time and increased from 1.03% at 0.05 µg/mL in 24 h to 69.23% at 500 µg/mL in 72 h. Therefore, only 30.77% of the cells were able to survive (viability%), which showed the high antitumor effect of green walnut husk extract. The amount of IC50 was also calculated, based on the drawing of the curve and the equation of the line, and it was equal to 94.80, 68.81, 28.21 µg/mL of husk over 24, 48 and 72 h, respectively. Vieira et al. [23] reported that, at 24 µg/mL concentration of hydroethanolic husk extract, the total HepG2 cell growth reduced by 50%. (GI_50_) [23]. Other studies have been conducted on the antitumor effect of pure extracted compounds from husks, such as juglone and terpenes, instead of the whole extract against various human cancer cell lines, including MCF-7, HCT-116, HeLa, K562, Raji, and THP-1 and high cytotoxic potential has been reported [6]. However, Soto-Maldonado et al. [26] stated that the toxicity of the final extract against the HL-60 cells was higher than that of the pure compound (juglone).

## 4. Conclusions

Although there has been increasing interest in finding naturally abundant sources of antioxidant compounds, a key challenge in securing potential sources is the absence of standardized protocols to measure the antioxidant power. In this study, for the first time, PAOT technology was used to investigate the antioxidant activity of one type of agricultural waste. The novelty of this study was the complete identification of the bioactive components and the use of the PAOT technique. Although the results obtained from the PAOT and DPPH methodologies show similar trends, the PAOT technique has the great advantage of producing data that can be reliably compared with other data, including when obtained from different studies. Thus, the green husks from seven walnut varieties were analyzed for their chemical compounds, their total phenolic content, as well as their antioxidant power. In total, 46 compounds were successfully identified from an apolar extract, while 11 compounds were characterized from a polar extract. The total content of phenolic compounds and the antioxidant power based on the dry weight of the husk showed strong potential. In this evaluation, the variety and color of the husk were effective parameters. Of the studied varieties, the differences in antioxidant power were probably due to the differences in the composition and content of the phenolics. The difference in the drying time of the husk in the Saman variety also caused an obvious difference in the antioxidant power, which was probably attributed to the oxidation and reduction of the compounds. Walnut husk has applications as traditional medicine for the treatment of skin diseases and the alleviation of pain. The findings of this study provide relatively complete information about the potential of walnut husk for its future application in food and pharmacological products. In conclusion, it can be stated that walnut husk without any purification process is a rich source of phenolics with powerful antioxidant properties. 

## Figures and Tables

**Figure 1 antioxidants-12-00052-f001:**
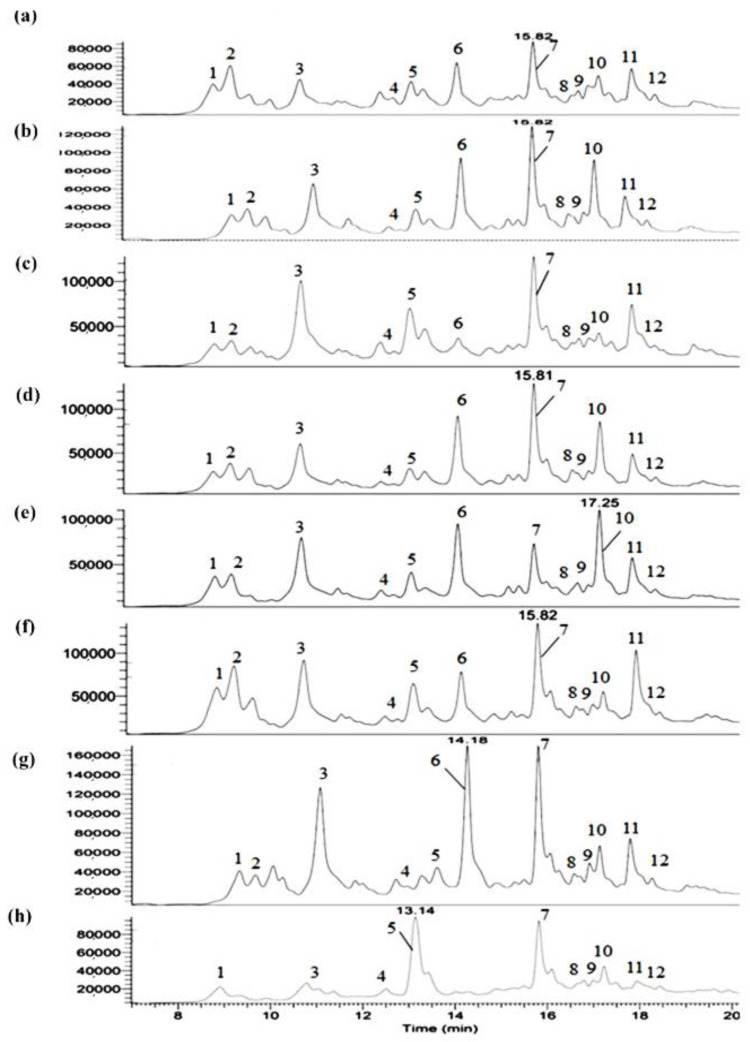
UV282 chromatograms of polar extracts of various varieties of walnut husk (**a**) Aytak; (**b**) Dena; (**c**) Chandler; (**d**) Raha3; (**e**) Raha 4; (**f**) Raha 6; (**g**) green Saman, and (**h**) brown Saman, obtained on the UHPLC-PDA-HRMS/MS system.

**Figure 2 antioxidants-12-00052-f002:**
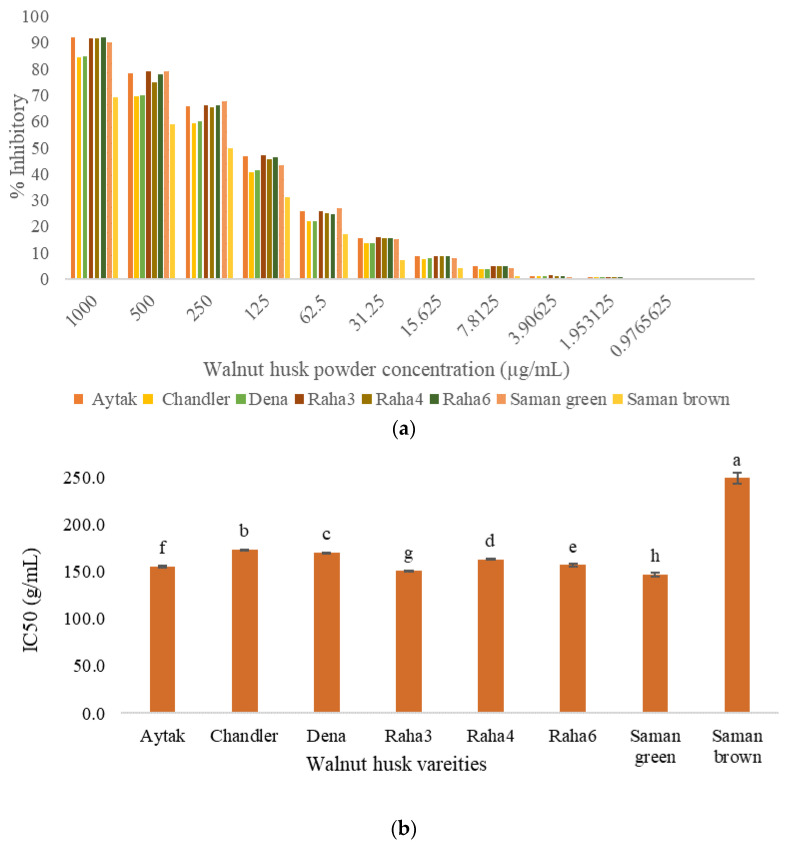
Antioxidant activity based on (**a**) the DPPH radical scavenging activity (inhibitory %) at different concentrations of husk powder, and on (**b**) the IC50 of various variants of walnut husk. Different letters in the same series indicate significant differences at the *p* < 0.05 level.

**Figure 3 antioxidants-12-00052-f003:**
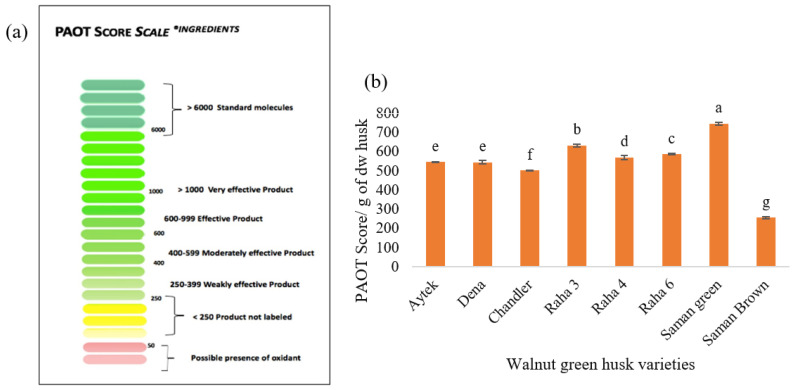
(**a**) PAOT liquid technology scale (ingredients), and (**b**) values of total antioxidant power of dried green walnut husk powder expressed with PAOT score/g of product.

**Figure 4 antioxidants-12-00052-f004:**
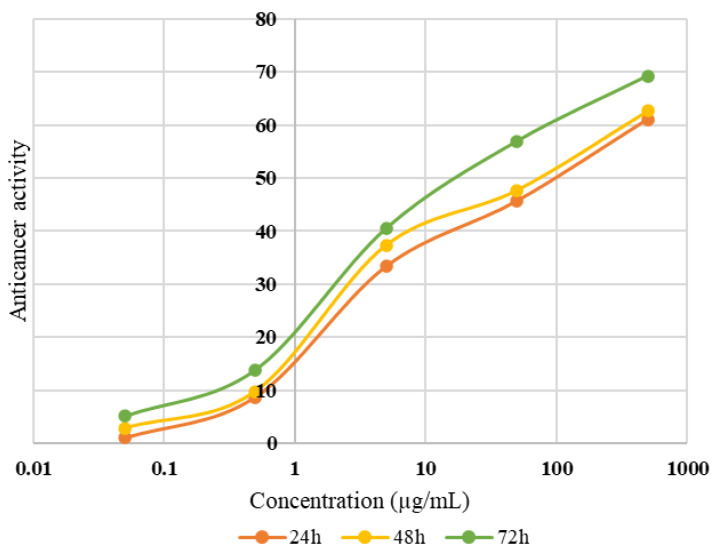
Anticancer activity of green walnut husk (Saman variety) against the human liver tumor cancer cell line (HepG2) over 24, 48, and 72 h.

**Table 1 antioxidants-12-00052-t001:** Chemical compounds identified in non-polar extracts of various varieties of walnut husk by gas chromatography-mass spectrometry (GC-MS).

Name		RT [min]	Raw Formula	Name	RT [min]	Raw Formula
Alkanes						
	Heptane, 2,3-dimethyl-	5.09	C_9_H_20_	Tetradecane	37.32	C_14_H_30_
	Octane, 4-methyl	5.36	C_9_H_20_	Pentadecane	43.39	C_15_H_32_
	Octane, 3-methyl	5.62	C_9_H_20_	Hexadecane	49.14	C_16_H_34_
	Nonane	6.69	C_9_H_20_	Octadecane	59.82	C_18_H_38_
	Octane, 2,5-dimethyl-	7.73	C_10_H_22_	Nonadecane	63.04	C_19_H_40_
	Octane, 2,6-dimethyl-	8.14	C_10_H_22_	Eicosane	67.76	C_20_H_42_
	Heptane, 3-ethyl-2-methyl-	8.46	C_10_H_22_	Heneicosane	69.70	C_21_H_44_
	Nonane, 4-methyl-	9.53	C_10_H_22_	Docosane	78.44	C_22_H_46_
	Nonane, 2-methyl-	9.67	C_10_H_22_	Tricosane	82.64	C_23_H_48_
	Nonane, 3-methyl-	10.0	C_10_H_22_	Tetracosane	86.66	C_2_4H_50_
	Decane	11.62	C_10_H_22_	Pentacosane	90.53	C_25_H_52_
	Undecane	17.78	C_11_H_24_	Hexacosane	94.27	C_26_H_54_
	Dodecane	24.40	C_12_H_26_	Heptacosane	97.88	C_27_H_56_
	Tridecane	30.99	C_13_H_28_	Octacosane	101.34	C_28_H_58_
Alkene						
	1,19-Eicosadiene	95.15	C_20_H_38_			
Naphthoquinone						
	Juglone	43.83	C_10_H_6_O_3_			
Terpenes						
	β-Limonene	13.83	C_10_H_16_	β-Tocopherol	109.31	C_28_H_48_O_2_
	Dl-Limonene	13.60	C_10_H_16_	γ-Tocopherol	109.77	C_28_H_48_O_2_
	Squalen	102.13	C_30_H_50_	α-Tocopherol	112.25	C_29_H_50_O_2_
Sterols						
	γ-Sitosterol	117.50	C_29_H_50_O	Lupeol	119.77	C_30_H_50_O
Fatty acids						
	Tetradecanoic acid,	34.87	C_15_H_30_O_2_	9. 12, 15-Octadecatrienoic acid (Z,Z,Z)-	84.21	C_19_H_32_
	Hexadecanoic acid,	69.36	C_17_H_34_O_2_	9-Octadecenoic acid (Z)	74.17	C_19_H_36_O_2_
	Octadecanoic acid	73.12	C_19_H_38_O_2_	9,12-Octadecadienoic acid (Z,Z)	77.12	C_19_H_34_O_2_
Other compounds						
	1H-Indene	7.42	C_9_H_16_	Phenol, 2,4-bis (1,1-dimethylethyl)	45.88	C_17_H_30_OSi

**Table 2 antioxidants-12-00052-t002:** The fatty acid profile of non-polar extracts of walnut husk.

Methylated Fatty Acid	RT [min]	Aytak	Chandler	Dena	Raha 3	Raha 4	Raha 6	Saman Green	Saman Brown
Myristic	8.40	2.2 ^bc^ ± 0.1	1.4 ^e^ ± 0.0	2.1 ^c^ ± 0.0	2.8 ^b^ ± 0.0	1.5 ^d^ ± 0.0	2.1 ^c^ ± 0.1	1.5 ^d^ ± 0.0	4.5 ^a^ ± 0.9
Palmitic	9.30	20.1 ^c^ ± 0.1	18.5 ^e^ ± 0.1	21.5 ^b^ ± 0.1	19.1 ^d^ ± 0.1	20.7 ^c^ ± 0.0	21.0 ^b^ ± 0.1	20.3 ^c^ ± 0.2	24.2 ^a^ ± 0.5
Stearic	10.70	8.9 ^c^ ± 0.1	10.0 ^b^ ± 0.2	8.7 ^c^ ± 0.1	8.7 ^c^ ± 0.4	9.0 ^c^ ± 0.2	9.7 ^b^ ± 0.3	9.5 ^bc^ ± 0.1	11.2 ^a^ ± 0.7
Oleic	11.17	47.4 ^c^ ± 0.4	50.3 ^a^ ± 0.2	49.1 ^b^ ± 0.1	47.1 ^c^ ± 0.1	49.2 ^b^ ± 0.3	46.4 ^d^ ± 0.4	40.2 ^e^ ± 0.6	39.1 ^f^ ± 0.8
Linoleic	11.96	15.3 ^d^ ± 0.1	18.5 ^b^ ± 0.2	17.4 ^c^ ± 0.1	14.4 ^e^ ± 0.3	17.9 ^c^ ± 0.1	19.0 ^b^ ± 0.2	23.5 ^a^ ± 0.2	18.7 ^b^ ± 0.1
Linolenic	12.98	0.8 ^d^ ± 0.0	1.4 ^c^ ± 0.0	1.2 ^cd^ ± 0.1	1.3 ^cd^ ± 0.0	1.7 ^bc^ ± 0.0	1.8 ^b^ ± 0.0	5.0 ^a^ ± 0.1	2.3 ^b^ ± 0.9
Eicosenoic	14.10	5.3 ^b^ ± 0.5	-	-	6.6 ^a^ ± 0.2	-	-	-	-

The values are expressed as a percentage of total fatty acids. Different letters in each row indicate significant differences (*p* < 0.05).

**Table 3 antioxidants-12-00052-t003:** Chemical compounds identified in polar extracts of various varieties of walnut husk by ultra-high performance liquid chromatography in-line with a photodiode array detector and an orbitrap mass spectrometer (UHPLC-HESI+-HRMS/MS).

No.	Name	RT [min]	Formula	Calc. M_W_	*m*/*z*	Ref.
1	Neochlorogenic acid	9.08	C_16_H_18_O_9_	354.095	355.103	[33,34]
2	Taxifolin 7-glucoside	9.45	C_21_H_24_O_11_	466.181	467.189	[35]
3	Cianidanol or (+)-catechin	11.77	C_15_H_14_O_6_	290.079	291.086	[33,34,35]
4	Minoxidil	13.30	C_9_H_15_N_5_O	209.128	210.135	[36]
5	(±)-(2E)-Abscisic acid	13.34	C_15_H_20_O_4_	264.136	265.143	[37]
6	Unknown	14.33	C_33_H_27_N_6_O_11_P	714.147	358.081	-
7	1-Salicylate glucuronide	15.99	C_13_H_14_O_9_	314.064	315.071	-
8	Myricetin	17.01	C_21_H_20_O_12_	464.095	465.102	[33,34]
9	Quercetin-4′-glucoside	17.18	C_21_H_20_O_12_	464.096	465.102	[38]
10	Taxifolin	17.40	C_15_H_12_O_7_	304.058	305.066	[33,34]
11	Gallic acid derivative	18.11	C_7_H_6_O_5_	490.110	491.118	[21]
12	Quercetin-3-O-pentoside	18.22	C_20_H_18_O_11_	434.085	435.092	[34,38]

**Table 4 antioxidants-12-00052-t004:** Antibacterial properties (MIC (mg/mL), MBC (mg/mL), MBC/MIC (-)) of various varieties of walnut husk extracts against *S.aureus*, *B. subtilis*, *E. coli*, and *P. aeruginosa* collection (*n* = 6).

Varieties	*S. aureus*			*B. subtilis*			*E. coli*			*P. aeruginosa*		
	MIC	MBC	MBC/MIC	MIC	MBC	MBC/MIC	MIC	MBC	MBC/MIC	MIC	MBC	MBC/MIC
Aytak	2	2	1	1	2	2	0.5	1	2	4	4	1
Chandler	3	3	1	2	4	2	1	2	2	5	5	1
Dena	2	2	1	1	2	2	0.5	1	2	4	4	1
Raha 3	2	2	1	1	2	2	0.5	1	2	4	4	1
Raha 4	3	3	1	2	4	2	1	2	2	5	5	1
Raha 6	2	2	1	1	2	2	0.5	1	2	4	4	1
Green Saman	2	2	1	1	2	2	0.5	1	2	4	4	1
Brown Saman	3	3	1	1	2	2	1	2	2	7	7	1
Streptomycin	0.03	0.03	1	0.00	0.00	1	0.02	0.02	1	0.03	0.03	1
Penicillin	0.02	0.02	1	0.00	0.01	2	0.03	0.03	1	0.06	0.13	2

## Data Availability

All data generated or analyzed during this study are included in this article.

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
