# Peer review of "Phytochemical Composition, Antimicrobial, Anticancer Properties, and Antioxidant Potential of Green Husk from Several Walnut Varieties (Juglans regia L.)"

_antioxidants, 2022, doi:10.3390/antiox12010052_

Round 1
Reviewer 1 Report
In the manuscript "Phytochemical composition, antimicrobial, anticancer properties and antioxidant potential of green husk from several walnut varieties (Juglans regia L.)" the introduction provides the necessary background, and the methods are rigorously described. The results are clearly described.
Here are my comments in this manuscript:
- In line 64 the comma after “the working electrode” must be removed
- In the caption of table 2, in my opinion, the sentence "The values are expressed as a percentage of total fatty acids" should be added.
- Specify which type of extract, polar, non-polar or both are tested in the different assays.
The research design is appropriate but it would be interesting to know if there are uses (for example traditional medicine, supplements, ..) of the walnut husk supported by these results.
Author Response
Comment 1. In the manuscript "Phytochemical composition, antimicrobial, anticancer properties and antioxidant potential of green husk from several walnut varieties (Juglans regia L.)" the introduction provides the necessary background, and the methods are rigorously described. The results are clearly described.
Reply: Thank you for your positive opinions to our submission. We appreciate the constructive feedback and have striven to incorporate that feedback into our article.
Comment 2. In line 64 the comma after “the working electrode” must be removed
Reply: Thank you for your attention. The correction was done.
Comment 3. In the caption of table 2, in my opinion, the sentence "The values are expressed as a percentage of total fatty acids" should be added.
Reply: According to honorable reviewer comment, the sentence "The values are expressed as a percentage of total fatty acids" was added to Table 2 caption.
Comment 4. Specify which type of extract, polar, non-polar or both are tested in the different assays.
Reply: Thank you for your attention. The type of extracted in all assays were added.
Comment 5. The research design is appropriate but it would be interesting to know if there are uses (for example traditional medicine, supplements, ..) of the walnut husk supported by these results.
Reply: We add more details about its uses. “Walnut husk has application as traditional medicine for the treatment of skin diseases and the alleviation of pain. The findings of this study provide relatively complete information about the potential of walnut husk for its future application in food and pharmacological products. In conclusion, it can be stated that the walnut husk without any purification process is a rich source of phenolics with powerful antioxidant properties.” The experiment related to its use deviated from the main goal of this article. However, we are agreeing with the honorable reviewer's opinion about investigating the uses of walnut husk. So, we will show in our next work, how to use and formulate walnut extract for use in the food industry.
Reviewer 2 Report
The subject of the manuscript fell within the general scope of the journal and the sound of the manuscript is scientific. The subject has original contribution and interesting results about the antioxidant capacity of walnut husk using the PAOT method. However, the manuscript present several details that need to be corrected.
Line 91…..the authors mention that they collected the fruit after the full ripening…..what parameters did they use to establish full ripening? could you explain, please?
Lines 91-100…..the authors do not specify how they selected the brown husks,... please include this information
Lines 105-107… the authors mention that for Saman variety, freeze-drying of husks was done before and after the discoloration from green to brown....how was the husks discoloration from green to brown carried out? could you explain, please?
Lines 111-112…..please write the bacteria name in italics
Line 132…..the authors mention that The extraction process was the same…… the same process as which one?...it's confusing...can you write it more clearly?
Lines 141, 158, 159…..please include the correct symbol of temperature
Line 154….please check and correct the format to cite the references in the text
Lines 167, 174…..in the flow rate, and in arbitrary units, please write the number 1 as superscript
Line 180……please write the full name of AGC and put into parenthesis AGC
Line 209….please check the format to report the concentration units of phenolic compounds, if it is mg equivalent of gallic acid per g of dw, or mg equivalent of gallic acid/g dw of husk powder, and use the same format in all the manuscript
Line 224….the authors mention that “The antioxidant power was measured in walnut green husk”, and what happen with the brown husks? was the antioxidant capacity not measured in the brown husks? Please check and correct it.
Lines 242-243…..please write the bacteria names in italics
Line 253….. please mention where did you get the cell lines
Lines 267-269…. The authors do not mention the experimental design used and the statistical analysis performed in this study. Authors should run an ANOVA to analyze the significance of the data obtained. Please include this information.
Lines 295- 297….. the title of the table 1 mentions the compounds detected in the green husk extracts, however the table also includes the brown husk, please review and correct
Line 333-334….. Table 2 shows the results of the fatty acid profile obtained in the husk of the different varieties evaluated, both green and brown husks, but the title of the table refers only to the compounds of the green husks. ..in addition, the letters that indicate statistical significance are not clearly seenl....... please review and correct
Line 348…… The title of figure 1 indicates that the results presented correspond to those obtained from the extracts of the green husk, but the figure includes the results of the green and brown husks... please review and correct the title of figure 1
Line 360…..In table 3, please correct the compounds name of Myricetin, and Quercetin-3-O-pentoside
Lines 373-374……please check the compounds name and the retention time, they did not coincide with the information included in table 3
Line 380…..the authors mention “the neutral loss of NL = 33”…..please delete “NL =”
Lines 385, 393-394…..the authors cite the reference Sheng et al., (2021), but in line 288, they use a different format to cite other reference…..please check the format to cite the references and use the same throughout the manuscript
Line 400…please check the retention time of the Abscisic acid, it did not coincide with that included in table 3
Lines 430-431…. according to figure 2b, Saman brown presented the lowest activity, and not Chandler....or does it only refer to the green husks?....explain
Line 445….. according to figure 2b, the IC50 results of the green and brown husks are presented, but the title only mentions the green ones....please review and correct
Lines 450-454…. the authors include information that is repeated in figure 3a....I suggest that it be written in a different way
Line 457…. this information does not match the data included in figure 3b. All the green husks presented values in the PAOT score above 400, which is described as a moderately effective product.....please check and correct
Line 460….. the authors mention that the walnut husk can certainly be classified as a strong antioxidant.....according to the results obtained in the present study, it is the green husk that presented the highest antioxidant capacity, so I suggest that it be specified as walnut green husk.
Line 466…..the authors mention ”Among the two varieties of Saman..”, but the authors did not evaluate two varieties of Saman, rather they included two types of husk (green and brown).....I suggest review and correct, and also use the word between instead of among.
Lines 476-478…… I think the discussion about the low antioxidant capacity that was observed in the brown huks should be improved. According to figure 1, Saman brown also has 1-Salicylate glucuronide, which is the glucuronide conjugate of salicylic acid, and salicylic acid also function as an antioxidant...there is also a clear difference in the peak area of catechin (with antioxidant capacity), which is very low in Saman brown.......please explain
Lines 497, 498….please correct E. coli, P. aeruginosa and write them in italics
Line 504…..please write the bacteria names in italics
Lines 505, 522…..please correct the bacteria names, S. aureus, B. subtilis, E. coli, P. aeruginosa, and write in italics
Line 507…..please delete the reference year
Lines 509-517…..please write the bacteria names in italics
Lines 513, 515….the cites Fernández-Agulló, et al., (2013), and Oliveira et al (2008) are not included in the reference list
Lines 533, 539….please delete the reference year
Line 541….according to the text in lines 524-540, the results presented in figure 4 correspond to the anticancer activity of the walnut husk of the Saman variety…..please correct the title of the figure. In addition, it is suggested to include the title of the X axis, and the title of the Y axis
Author Response
Comment 1. The subject of the manuscript fell within the general scope of the journal and the sound of the manuscript is scientific. The subject has original contribution and interesting results about the antioxidant capacity of walnut husk using the PAOT method. However, the manuscript present several details that need to be corrected.
Reply: We thank you for your careful read and thoughtful comment on the manuscript antioxidants-2086144. We have carefully taken your comment into consideration in preparing our revision, which has resulted in a paper that is clearer and more compelling.
Comment 2. Line 91…..the authors mention that they collected the fruit after the full ripening…..what parameters did they use to establish full ripening? could you explain, please?
Reply: Empirically, when the walnut fruits are fully ripe, the color of the walnut kernel should be light and the inner kernels should be formed. The membrane between the two halves must be brown. Unripe walnuts are difficult to separate from the green husk.
Hulling rate, which represents the ripening stage of walnut fruits, was measured. Only the green husks that were completely separated from the walnut fruit were considered “hulling” and those green husks incompletely removed were considered “non-hulling”. Hulling rate was calculated according to the following equation (1):
Hulling rate %= (Ao/A) * 100
In this method, 10 kg of walnut fruits are collected and the more fruits that are completely separated from the husk showed the hulling rate according to the equation. The hulling rate of the samples was over 80%, which shows the full maturation of the fruits.
Comment 3. Lines 91-100…..the authors do not specify how they selected the brown husks,... please include this information
Reply: Thank you for your attention. The information was added.
Comment 4. Lines 105-107… the authors mention that for Saman variety, freeze-drying of husks was done before and after the discoloration from green to brown....how was the husks discoloration from green to brown carried out? could you explain, please?
Reply: The information was added. To obtain the brown type of Saman, fresh husk was stored in the refrigerator (4 ℃) for 2 weeks.
Comment 5. Lines 111-112…..please write the bacteria name in italics
Reply: We apologize for the typical errors in the manuscript. All of errors were corrected.
Comment 6. Line 132…..the authors mention that The extraction process was the same…… the same process as which one?...it's confusing...can you write it more clearly?
Reply: The extraction process steps of polar and apolar compounds were the same… Yes, we added more details to make it clear for readers.
Comment 7. Lines 141, 158, 159…..please include the correct symbol of temperature.
Reply: We apologize for the typical errors in the manuscript. All of errors were corrected.
Comment 8. Line 154….please check and correct the format to cite the references in the text
Reply: Thank you for your attention. The year was deleted.
Comment 9. Lines 167, 174…..in the flow rate, and in arbitrary units, please write the number 1 as superscript
Reply: All corrections were done.
Comment 10. Line 180……please write the full name of AGC and put into parenthesis AGC
Reply: Thank you for your attention. The full name “Automatic gain control (AGC) target” was added.
Comment 11. Line 209….please check the format to report the concentration units of phenolic compounds, if it is mg equivalent of gallic acid per g of dw, or mg equivalent of gallic acid/g dw of husk powder, and use the same format in all the manuscript
Reply: Thank you for your attention. The correction was done.
Comment 12. Line 224….the authors mention that “The antioxidant power was measured in walnut green husk”, and what happen with the brown husks? was the antioxidant capacity not measured in the brown husks? Please check and correct it.
Reply: The honorable reviewer opinion is completely true. The antioxidant capacity was measured for both types (brown and green). We correct it with walnut husk.
Comment 13. Lines 242-243…..please write the bacteria names in italics
Reply: Thank you for your attention. All corrections were done.
Comment 14. Line 253….. please mention where did you get the cell lines
Reply: Thank you for your attention. The information was added.
Comment 15. Lines 267-269…. The authors do not mention the experimental design used and the statistical analysis performed in this study. Authors should run an ANOVA to analyze the significance of the data obtained. Please include this information.
Reply: The honorable reviewer opinion is completely true. The information of experimental design was added.
Comment 16. Lines 295- 297….. the title of the table 1 mentions the compounds detected in the green husk extracts, however the table also includes the brown husk, please review and correct
Reply: The honorable reviewer opinion is completely true. The correction was done.
Comment 17. Line 333-334….. Table 2 shows the results of the fatty acid profile obtained in the husk of the different varieties evaluated, both green and brown husks, but the title of the table refers only to the compounds of the green husks. ..in addition, the letters that indicate statistical significance are not clearly seenl....... please review and correct
Reply: Thank you for your attention. All corrections were done.
Comment 18. Line 348…… The title of figure 1 indicates that the results presented correspond to those obtained from the extracts of the green husk, but the figure includes the results of the green and brown husks... please review and correct the title of figure 1
Reply: Thank you for your attention. The correction was done.
Comment 19. Line 360…..In table 3, please correct the compounds name of Myricetin, and Quercetin-3-O-pentoside
Reply: Thank you for your attention. All corrections were done.
Comment 20. Lines 373-374……please check the compounds name and the retention time, they did not coincide with the information included in table 3
Reply: Thank you for your attention. All corrections were done.
Comment 21. Line 380…..the authors mention “the neutral loss of NL = 33”…..please delete “NL =”
Reply: Thank you for your attention. All corrections were done.
Comment 22. Lines 385, 393-394…..the authors cite the reference Sheng et al., (2021), but in line 288, they use a different format to cite other reference…..please check the format to cite the references and use the same throughout the manuscript
Reply: Thank you for your attention. All corrections for the references were done.
Comment 23. Line 400…please check the retention time of the Abscisic acid, it did not coincide with that included in table 3
Reply: The correction was done.
Comment 24. Lines 430-431…. according to figure 2b, Saman brown presented the lowest activity, and not Chandler....or does it only refer to the green husks?....explain
Reply: The honorable reviewer opinion is completely true. It only refers to green husks.
Comment 25. Line 445….. according to figure 2b, the IC50 results of the green and brown husks are presented, but the title only mentions the green ones....please review and correct
Reply: The correction was done.
Comment 26. Lines 450-454…. the authors include information that is repeated in figure 3a....I suggest that it be written in a different way
Reply: The honorable reviewer opinion is completely true. The correction was done.
Comment 27. Line 457…. this information does not match the data included in figure 3b. All the green husks presented values in the PAOT score above 400, which is described as a moderately effective product.....please check and correct
Reply: Samples with a value comprised between 600 and 999 are considered as effective products, so Saman green (746.88) and Raha 3 (633.33) are rated as effective.
Comment 28. Line 460….. the authors mention that the walnut husk can certainly be classified as a strong antioxidant.....according to the results obtained in the present study, it is the green husk that presented the highest antioxidant capacity, so I suggest that it be specified as walnut green husk.
Reply: The honorable reviewer opinion is completely true. The correction was done
Comment 29. Line 466…..the authors mention ”Among the two varieties of Saman..”, but the authors did not evaluate two varieties of Saman, rather they included two types of husk (green and brown).....I suggest review and correct, and also use the word between instead of among.
Reply: The honorable reviewer opinion is completely true. The corrections were done
Comment 30. Lines 476-478…… I think the discussion about the low antioxidant capacity that was observed in the brown huks should be improved. According to figure 1, Saman brown also has 1-Salicylate glucuronide, which is the glucuronide conjugate of salicylic acid, and salicylic acid also function as an antioxidant...there is also a clear difference in the peak area of catechin (with antioxidant capacity), which is very low in Saman brown.......please explain
Reply: Thanks for reviewer positive suggestion. The more explanation was added.
Comment 31. Lines 497, 498….please correct E. coli, P. aeruginosa and write them in italics
Reply: Thank you for your attention. All corrections were done.
Comment 32. Line 504…..please write the bacteria names in italics
Reply: All corrections were done.
Comment 33. Lines 505, 522…..please correct the bacteria names, S. aureus, B. subtilis, E. coli, P. aeruginosa, and write in italics
Reply: Thank you for your attention. All corrections were done.
Comment 34. Line 507…..please delete the reference year
Reply: Thank you for your attention. All corrections were done.
Comment 35. Lines 509-517…..please write the bacteria names in italics
Reply: All corrections were done.
Comment 36. Lines 513, 515….the cites Fernández-Agulló, et al., (2013), and Oliveira et al (2008) are not included in the reference list
Reply: Thank you for your attention. The correction was done.
Comment 37. Lines 533, 539….please delete the reference year
Reply: The correction was done.
Comment 38. Line 541….according to the text in lines 524-540, the results presented in figure 4 correspond to the anticancer activity of the walnut husk of the Saman variety…..please correct the title of the figure. In addition, it is suggested to include the title of the X axis, and the title of the Y axis
Reply: Thank you for your attention. All corrections were done.